# Evaluation of Neural Architectures Trained with Square Loss vs Cross-Entropy in Classification Tasks

**Like Hui**
Computer Science and Engineering
University of California, San Diego
San Diego, CA 92093
lhui@ucsd.edu

**Mikhail Belkin**
Halıcıoğlu Data Science Institute
University of California, San Diego
San Diego, CA 92093
mbelkin@ucsd.edu

## Abstract

Modern neural architectures for classification tasks are trained using the cross-entropy loss, which is widely believed to be empirically superior to the square loss. In this work we provide evidence indicating that this belief may not be well-founded. We explore several major neural architectures and a range of standard benchmark datasets for NLP, automatic speech recognition (ASR) and computer vision tasks to show that these architectures, with the same hyper-parameter settings as reported in the literature, perform comparably or better when trained with the square loss, even after equalizing computational resources. Indeed, we observe that the square loss produces better results in the dominant majority of NLP and ASR experiments. Cross-entropy appears to have a slight edge on computer vision tasks.

We argue that there is little compelling empirical or theoretical evidence indicating a clear-cut advantage to the cross-entropy loss. Indeed, in our experiments, performance on nearly all non-vision tasks can be improved, sometimes significantly, by switching to the square loss. Furthermore, training with square loss appears to be less sensitive to the randomness in initialization. We posit that training using the square loss for classification needs to be a part of best practices of modern deep learning on equal footing with cross-entropy.

## 1 Introduction

Modern deep neural networks are nearly universally trained with cross-entropy loss in classification tasks. To illustrate, cross-entropy is the only loss function specifically discussed in connection with training neural networks for classification in popular references (Goodfellow et al., 2016; Zhang et al., 2020). It is the default for classification in widely used packages such as NLP implementation Hugging Face Transformers (Wolf et al., 2019), speech classification by ESPnet (Watanabe et al., 2018) and image classification implemented by torchvision (Marcel & Rodriguez, 2010). Yet we know of few empirical evaluations or compelling theoretical analyses to justify the predominance of cross-entropy in practice. In what follows, we use a number of modern deep learning architectures and standard datasets across the range of tasks of natural language processing, speech recognition and computer vision domains as a basis for a systematic comparison between the cross-entropy and square losses. The square loss (also known as the Brier score (Brier, 1950) in the classification context) is a particularly useful basis for comparison since it is nearly universally used for regression tasks and is available in all major software packages. To ensure a fair evaluation, for the square loss we use hyper-parameter settings and architectures exactly as reported in the literature for cross-entropy, with the exception of the learning rate, which needs to be increased in comparison with cross-entropy and, for problems with a large number of classes (42 or more in our experiments), loss function rescaling (see Section 5).

Our evaluation includes 20 separate learning tasks[1] (neural model/dataset combinations) evaluated in terms of the error rate or, equivalently, accuracy (depending on the prevalent domain conventions). We also provide some additional domain-specific evaluation metrics – F1 for NLP tasks, and Top-5 accuracy for ImageNet. Training with the square loss provides accuracy better or equal to that of cross-entropy in 17 out of 20 tasks. These results are for averages over multiple random initalizations, results for each individual initialization are similar. Furthermore, we find that training with the square loss has smaller variance with respect to the randomness of the initialization in the majority of our experiments.

Our results indicate that the models trained using the square loss are not just competitive with same models trained with cross-entropy across nearly all tasks and settings but, indeed, provide better classification results in the majority of our experiments. The performance advantage persists even when we equalize the amount of computation by choosing the number of epochs for training the square loss to be the same as the optimal (based on validation) number of epochs for cross-entropy, a setting favorable to cross-entropy.

Note that with the exception of the learning rate, we utilized hyper-parameters reported in the literature, originally optimized for the cross-entropy loss. This suggests that further improvements in performance for the square loss can potentially be obtained by hyper-parameter tuning.

Based on our results, we believe that the performance of modern architectures on a range of classification tasks may be improved by using the square loss in training. We conclude that the choice between the cross-entropy and the square loss for training needs to be an important aspect of model selection, in addition to the standard considerations of optimization methods and hyper-parameter tuning.

**A historical note.** The modern ubiquity of cross-entropy loss is reminiscent of the predominance of the hinge loss in the era of the Support Vector Machines (SVM). At the time, the prevailing intuition had been that the hinge loss was preferable to the square loss for training classifiers. Yet, the empirical evidence had been decidedly mixed. In his remarkable thesis (Rifkin, 2002), Ryan Rifkin conducted an extensive empirical evaluation and concluded that *"the performance of the RLSC* [square loss] *is essentially equivalent to that of the SVM* [hinge loss] *across a wide range of problems, and the choice between the two should be based on computational tractability considerations"*. More recently, the experimental results in (Que & Belkin, 2016) show an advantage to training with the square loss over the hinge loss across the majority of the tasks, paralleling our results in this paper. We note that conceptual or historical reasons for the current prevalence of cross-entropy in training neural networks are not entirely clear.

**Theoretical considerations.** The accepted justification of cross-entropy and hinge loss for classification is that they are better "surrogates" for the 0-1 classification loss than the square loss, e.g. (Goodfellow et al., 2016), Section 8.1.2. There is little theoretical analysis supporting this point of view. To the contrary, the recent work (Muthukumar et al., 2020) proves that in certain over-parameterized regimes, the classifiers obtained by minimizing the hinge loss and the square loss in fact the same. While the hinge loss is different from cross-entropy, these losses are closely related in certain settings (Ji & Telgarsky, 2019; Soudry et al., 2018). See (Muthukumar et al., 2020) for a more in-depth theoretical discussion of loss functions and the related literature.

**Probability interpretation of neural network output and calibration.** An argument for using the cross-entropy loss function is sometimes based on the idea that networks trained with cross-entropy are able to output probability of a new data point belonging to a given class. For linear models in the classical analysis of logistic regression, minimizing cross-entropy (logistic loss) indeed yields the maximum likelihood estimator for the model (e.g.,(Harrell Jr, 2015), Section 10.5). Yet, the relevance of that analysis to modern highly non-linear and often over-parameterized neural networks is questionable. For example, in (Gal & Ghahramani, 2016) the authors state that *"In classification, predictive probabilities obtained at the end of the pipeline (the softmax output) are often erroneously interpreted as model confidence"*. Similarly, the work (Xing et al., 2019) asserts that *"for DNNs with conventional (also referred as 'vanilla') training to minimize the softmax cross-entropy loss, the outputs do not contain sufficient information for well-calibrated confidence estimation"*. Thus, accurate class probability estimation cannot be considered an unambiguous advantage

---

[1]We note WSJ and Librispeech datasets have two separate classification tasks in terms of the evaluation metrics, based on the same learned acoustic model. We choose to count them as separate tasks.

of neural networks trained with cross-entropy. While the analysis of calibration for different loss functions is beyond the scope of this paper, we note that in many practical settings accurate classification, the primary evaluation metric of this work, takes precedence over the probability estimation.

**Domain applicability.** It is interesting to note that in our experiments the square loss generally performs better on NLP and ASR tasks, while cross-entropy has a slight edge on computer vision. It is tempting to infer that the square loss is suitable for NLP and speech, while cross-entropy may be more appropriate for training vision architectures. Yet we are wary of over-interpreting the evidence. In particular, we observe that the cross-entropy has a significant performance advantage on just a single vision architecture (EfficientNet (Tan & Le, 2019) trained on ImageNet). The rest of the vision results are quite similar between square loss and cross-entropy and are likely to be sensitive to the specifics of optimization and parameter tuning. Understanding whether specific loss functions are better suited for certain domain will require more in-depth experimental work.

**Related work.** The choice of a loss function is an integral and essential aspect of training neural networks. Yet we are aware of few comparative analyses of loss functions and no other systematic studies of modern architectures across a range of datasets. Kline & Berardi (2005) compared the effectiveness of squared-error versus cross-entropy in estimating posterior probabilities with small neural networks, five or less nodes in each layer, and argued that cross-entropy had a performance advantage. Golik et al. (2013) provided a comparison of cross-entropy and squared error training for a hybrid HMM/neural net model for one ASR and one handwriting recognition datasets. The authors observed that with a good initialization by pre-training, training with the squared error had better performance than the cross-entropy. Sangari & Sethares (2015) analyzed the convergence of mean squared error (MSE) and cross-entropy under the normalized logistic regression model (Soft-Max) setting, and indicated the MSE loss function is robust to the true model parameter values and can converge to the same parameter estimation variance of the cross-entropy loss function with half the number of gradient descent iterations. Janocha & Czarnecki (2017) compared several different loss functions on MNIST and CIFAR-10 datasets concluding that *"depending on the application of the deep model – losses other than log loss* [cross-entropy] *are preferable"*. A recent work (Demirkaya et al., 2020) provided a theoretical comparison of square and cross-entropy losses for training mixture models. The authors argued that the cross-entropy loss has more favorable optimization landscapes in multiclass settings. To alleviate that issue, they proposed rescaling of the loss function equivalent to choosing parameter $k$ in Section 5. The authors showed that rescaling allowed the square loss to become competitive with cross-entropy on CIFAR-100, a finding that aligns with the results in our paper.

## 2 EXPERIMENTS

We conducted experiments on a number of benchmark datasets for NLP, ASR and computer vision, following the standard recipes given in recent papers of each domain. Four NLP datasets are MRPC, SST-2, QNLI and QQP. TIMIT, WSJ and Librispeech are three standard datasets used for training ASR systems. For vision experiments, we choose MNIST, CIFAR-10, and ImageNet. To the best of our knowledge, we are the first to experimentally compare the square loss and the cross-entropy on a wide range of datasets with different size, dimensionality (number of features) and the number of classes (up to 1000 class numbers). See Appendix A for references and description.

**Architectures.** In what follows we explore several widely used modern neural architectures. For NLP tasks, we implement classifiers with a fine-tuned BERT (Devlin et al., 2018), a LSTM+Attention model (Chen et al., 2017), and a LSTM+CNN model (He & Lin, 2016). Joint CTC-Attention based model (Kim et al., 2017), triggered attention model with VGG and BLSTM modules (Moritz et al., 2019) are used for ASR tasks. Note that for the CTC-Attention based model, the original loss function is a weighted sum of the cross-entropy and the CTC loss. When training with the square loss, we only replace the cross-entropy to be the square loss, and keep the CTC loss untouched. For vision tasks, we use TCNN (Bai et al., 2018), Wide ResNet (Zagoruyko & Komodakis, 2016), ResNet (He et al., 2016) and EfficientNet (Tan & Le, 2019) architectures.

**Experimental protocols.** For training with the cross-entropy loss, we use a standard protocol, which is to stop training after the validation accuracy does not improve for five consecutive epochs. For the square loss we use two protocols. The first one is the same as for cross-entropy. The second protocol is to train the square loss using the number of epochs selected when training the cross-entropy loss with the first protocol. The second protocol is designed to equalize the usage of computational resources between the square loss and cross-entropy and is favorable to cross-entropy.

Following the hyper-parameter settings of the architectures in the literature, we re-implement the models trained with the cross-entropy loss keeping the same architecture and hyper-parameter settings. We train the same models using the square loss, employing our two experimental protocols. The only alteration to the parameters of the network reported in the literature is adjustment of the learning rate. For datasets with a large number of labels (42 or more in our experiments) we apply loss function rescaling (see Section 5).

The key points for the implementation are described in Section 5. The implementation details and specific hyper-parameter settings are given in Appendix B. See Appendix D for a summary of comparisons between the original results and our re-implementations. Additionally, we report the results on validation sets and training sets in Appendix C.

The results presented below are average results of 5 runs corresponding to 5 different random initalizations for each task. The result across initializations are given in Section 3.

## 2.1 NLP EXPERIMENTS

We conduct 2-class classification tasks from NLP domain. The datasets information is summarized in Table 1. As in (Wang et al., 2018), we report accuracy and F1 scores for MRPC and QQP datasets, and report accuracy for SST-2 and QNLI.

Table 1: NLP task statistics and descriptions

| Corpus | |Train| | |Test| | #classes | Metric | Domain |
|---|---|---|---|---|---|
| MRPC (Dolan & Brockett, 2005) | 3.7K | 1.7K | 2 | acc./F1 | news |
| SST-2 (Socher et al., 2013) | 67K | 1.8K | 2 | acc. | movie reviews |
| QNLI (Rajpurkar et al., 2016) | 105K | 5.4K | 2 | acc. | Wikipedia |
| QQP (Iyer et al., 2017) | 364K | 391K | 2 | acc./F1 | social QA questions |

Table 2 gives the accuracy and Table 3 gives the F1 scores of the neural models on NLP tasks. As

Table 2: NLP results, accuracy

| Model | Task | train with square loss (%) | train with cross-entropy (%) | square loss w/ same epochs as CE (%) |
|---|---|---|---|---|
| BERT (Devlin et al., 2018) | MRPC | **83.8** | 82.1 | 83.6 |
| | SST-2 | **94.0** | 93.9 | 93.9 |
| | QNLI | **90.6** | **90.6** | **90.6** |
| | QQP | **88.9** | **88.9** | 88.8 |
| LSTM+Attention (Chen et al., 2017) | MRPC | **71.7** | 70.9 | 71.5 |
| | QNLI | **79.3** | 79.0 | **79.3** |
| | QQP | **83.4** | 83.1 | **83.4** |
| LSTM+CNN (He & Lin, 2016) | MRPC | **73.2** | 69.4 | 72.5 |
| | QNLI | **76.0** | **76.0** | **76.0** |
| | QQP | 84.3 | **84.4** | 84.3 |

can be seen in Table 2, in 9 out of 10 tasks using the square loss has better/equal accuracy compared with using the cross-entropy, and in terms of F1 score (see Table 3), 5 out of 6 tasks training with the square loss outperform training with the cross-entropy loss. Even with same epochs, i.e. with same computation cost, using the square loss has equal/better accuracy in 8 out of 10 tasks , and has higher F1 score in 5 out of 6 tasks.

Table 3: NLP results, F1 scores

| Model | Task | train with square loss (%) | train with cross-entropy (%) | square loss w/ same epochs as CE (%) |
|---|---|---|---|---|
| BERT (Devlin et al., 2018) | MRPC | **88.1** | 86.7 | 88.0 |
| | QQP | **70.9** | 70.7 | 70.7 |
| LSTM+Attention (Chen et al., 2017) | MRPC | **80.9** | 80.6 | 80.7 |
| | QQP | **62.6** | 62.3 | **62.6** |
| LSTM+CNN (He & Lin, 2016) | MRPC | **81.0** | 78.2 | **81.0** |
| | QQP | 60.3 | **60.5** | 60.3 |

We observe the relative improvements brought by training with the square loss vary with different model architectures, and other than LSTM+CNN model on QQP dataset, all architectures trained

with the square loss have better/equal accuracy and F1 score. The performance of loss functions also varies with data size, especially for MRPC, which is a relatively small dataset, all model architectures trained with the square loss gives significantly better results than the cross-entropy.

## 2.2 Automatic Speech Recognition (ASR) experiments

We consider three datasets, TIMIT, WSJ and Librispeech, and all are ASR tasks. For Librispeech, we choose its train-clean-100 as training set, dev-clean and test-clean as validation and test set. We report phone error rate (PER) and character error rate (CER) for TIMIT, word error rate (WER) and CER for both WSJ and Librispeech. A brief description of the datasets used in our ASR experiments is given in Table 4[2]. Note that we only alter the training loss of the acoustic model, while keeping

Table 4: ASR task statistics and descriptions

| Corpus | |Train| | |Test| | #classes | Metric | Domain |
|---|---|---|---|---|---|
| TIMIT (Garofolo et al., 1993) | 1.15M | 54K | 42 / 27 | PER / CER | 3.2 hours (training set) telephone English |
| WSJ (Paul & Baker, 1992) | 28.8M | 252K | 52* | WER / CER | 80 hours (training set) read newspapers |
| Librispeech (Panayotov et al., 2015) | 36M | 1M | 1000* | WER / CER | 100 hours (training set) audio books |

\* This is the number of classes used for training the acoustic model.

the language model and decoding part the same as described in the literature. The acoustic model is a classifier with the dictionary size as the class number. For TIMIT, getting PER and CER needs two different acoustic models, i.e. they are two separate classification tasks, 42-class classification for PER, and 27-class classification for CER. For WSJ, the size of dictionary used for acoustic model is 52. WER and CER of WSJ are calculated with one acoustic model. Hence for WSJ it is a 52-class classification task for both WER and CER. Acoustic model of Librispeech is a 1000-class classifier for both WER and CER, as we use 1000 unigram (Jurafsky, 2000) based dictionary. The results are in Table 5.

Table 5: ASR results, error rate

| Model | Task | train with square loss (%) | train with cross-entropy (%) | square loss w/ same epochs as CE (%) |
|---|---|---|---|---|
| Attention+CTC | TIMIT (PER) | **20.8** | 20.8 | **20.8** |
| (Kim et al., 2017) | TIMIT (CER) | **32.5** | 33.4 | **32.5** |
| VGG+BLSTMP | WSJ (WER) | **5.1** | 5.3 | **5.1** |
| (Moritz et al., 2019) | WSJ (CER) | **2.4** | 2.5 | **2.4** |
| VGG+BLSTM | Librispeech (WER) | **9.8** | 10.6 | 10.3 |
| (Moritz et al., 2019) | Librispeech (CER) | **9.7** | 10.7 | 10.2 |

We see that the square loss performs better (equal for TIMIT PER result) in all of our tasks. It is interesting to observe that the performance advantage of the square loss reported in Table 5 increases with dataset size. In particular, the relative advantage of the square loss (9.3% relative improvement on CER, and 7.5% on WER, respectively) is largest for the biggest dataset, Librispeech. On WSJ, using the square loss has ~4% relative improvement on both CER and WER, while the results on TIMIT for the square loss and cross-entropy are very similar. The question of whether this dependence between the data size and the relative advantage of the square loss over cross-entropy is a coincidence or a recurring pattern requires further investigation.

For TIMIT and WSJ, we observed that training with both the square loss and the cross-entropy need same epochs to converge. The two training protocols for training with the square loss have same performance, and both are comparable/better than training with the cross-entropy. On Librispeech, the square loss needs more epochs, but provides better performance.

## 2.3 Computer vision experiments

For vision tasks we conduct experiments on MNIST, CIFAR-10 and ImageNet, as in Table 6.

---

[2]We measure the data size in terms of frame numbers, i.e. data samples. As we take frame shift to be 10ms, 1 hour data ∼ 360k frames.

Table 6: Vision task statistics and descriptions

| Corpus | |Train| | |Test| | #classes | Metric | Domain |
|---|---|---|---|---|---|
| MNIST (LeCun et al., 1998) | 60K | 10K | 10 | acc. | $28 \times 28$ |
| CIFAR-10 (Krizhevsky & Hinton, 2009) | 50K | 10K | 10 | acc. | $32 \times 32$ |
| ImageNet (Russakovsky et al., 2015) | ~1.28M | 50K[3] | 1000 | acc. Top-5 acc. | $224 \times 224$ |

As in Table 7, on MNIST and CIFAR-10, training with the square loss and the cross-entropy have comparable accuracy. On much larger ImageNet, with ResNet-50 architecture, the accuracy and Top-5 accuracy of using the square loss are comparable with the ones got by using the cross-entropy loss. While with EfficientNet, using the cross-entropy shows better results. The performance of different loss functions varies among different architectures. On MNIST and CIFAR-10, we use exactly the same hyper-parameters well-selected for the cross-entropy loss. For ImageNet, we adjust the learning rate and add a simple rescaling scheme (see Section 5), all other hyper-parameters are the same as for the cross-entropy loss. The performance of using the square loss can improve with more hyper-parameter tuning.

Table 7: Vision results, accuracy

| Model | Task | train with square loss (%) | train with cross-entropy (%) | square loss w/ same epochs as CE (%) |
|---|---|---|---|---|
| TCNN (Bai et al., 2018) | MNIST (acc.) | **97.7** | **97.7** | **97.7** |
| W-Resnet (Zagoruyko & Komodakis, 2016) | CIFAR-10 (acc.) | 95.9 | **96.3** | 95.9 |
| ResNet-50 | ImageNet (acc.) | **76.2** | 76.1 | 76.0 |
| (He et al., 2016) | ImageNet (Top-5 acc.) | **93.0** | **93.0** | 92.9 |
| EfficientNet | ImageNet (acc.) | 74.6 | **77.0** | 74.6 |
| (Tan & Le, 2019) | ImageNet (Top-5 acc.) | 92.7 | **93.3** | 92.7 |

For all three datasets, training with the square loss converges as fast as training with the cross-entropy, and our two experimental protocols for the square loss result in same accuracy performance (except ImageNet with ResNet-50 model).

## 3 PERFORMANCE ACROSS DIFFERENT INITIALIZATIONS

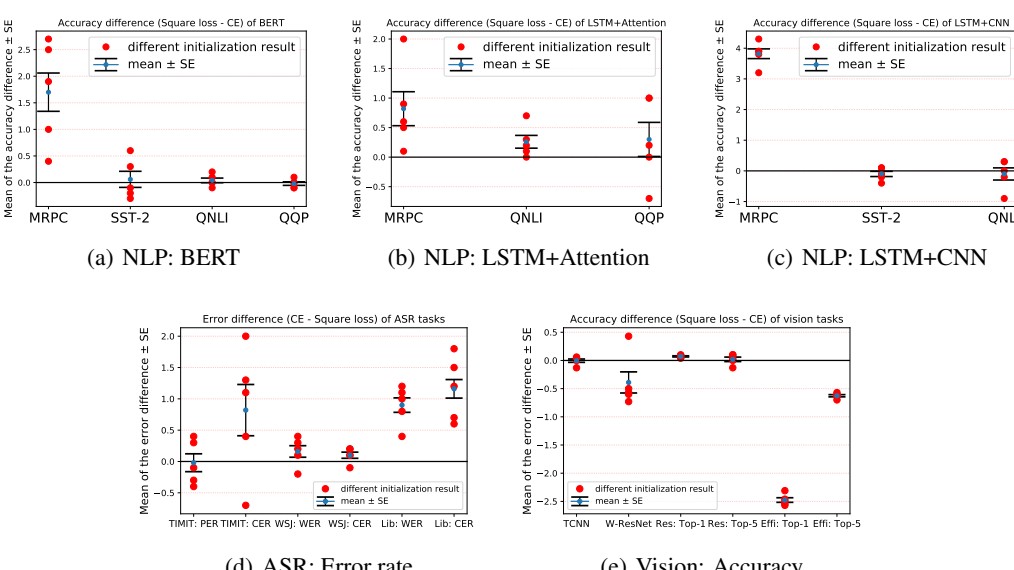

(a) NLP: BERT  (b) NLP: LSTM+Attention  (c) NLP: LSTM+CNN

(d) ASR: Error rate  (e) Vision: Accuracy

Figure 1: Difference between accuracy (or error rate) between square loss and CE for each initialization. (Square loss acc. - CE acc.) is shown for accuracy, (CE - Square loss) for error rate.

[3]We report validation set size, as results are on validation, following the papers for ImageNet tasks.

To evaluate the stability of the results with respect to the randomness of model initialization we analyze the results for each random seed initialization. For each random seed, we calculate *the difference* between the the accuracy (or the error) of networks trained with the square loss and the cross-entropy respectively. We present the results with error bars for one standard deviation in Figure 1. Absolute error and accuracy results for each run and the corresponding standard deviations are given in Appendix F.

Table 8 (Libri is short for Librispeech and I-Net is short for ImageNet) shows the standard deviation of test accuracy/error for training with the square loss and cross-entropy. Square loss has smaller variance in 15 out of 20 tasks, which indicates that training with the square loss is less sensitive to the randomness in the training process.

Table 8: Standard deviation of test accuracy/error. Smaller number is bolded.

| Model | Dataset | Square loss | CE |
|---|---|---|---|
| BERT | MRPC | **0.484** | 0.766 |
| | SST-2 | 0.279 | **0.173** |
| | QNLI | 0.241 | **0.205** |
| | QQP | **0.045** | 0.063 |
| LSTM +Attention | MRPC | **0.484** | 0.786 |
| | QNLI | **0.210** | 0.371 |
| | QQP | 0.566 | **0.352** |
| LSTM +CNN | MRPC | **0.322** | 0.383 |
| | QNLI | **0.173** | 0.286 |
| | QQP | 0.458 | **0.161** |
| Attention +CTC | TIMIT (PER) | 0.508 | **0.249** |
| | TIMIT (CER) | **0.361** | 0.873 |
| VGG+ BLSTMP | WSJ (WER) | **0.184** | 0.249 |
| | WSJ (CER) | **0.077** | 0.118 |
| VGG+ BLSTM | Libri (WER) | **0.126** | 0.257 |
| | Libri (CER) | **0.148** | 0.316 |
| TCNN | MNIST | **0.161** | 0.173 |
| W-ResNet | CIFAR-10 | **0.184** | 0.481 |
| ResNet-50 | I-Net (Top-1) | **0.032** | 0.045 |
| | I-Net (Top-5) | 0.126 | **0.045** |
| EfficientNet | I-Net (Top-1) | 0.138 | **0.122** |
| | I-Net (Top-5) | **0.089** | **0.089** |

## 4 OBSERVATIONS DURING TRAINING

There are several interesting observations in terms of the optimization speed comparing training with the square loss and the cross-entropy loss. We give the experimental observations for the cases when the class number is small, as for our NLP tasks, which are all 2-class classification tasks, and when the class number is relatively large, as for Libripseech and ImageNet (both have 1000 classes).

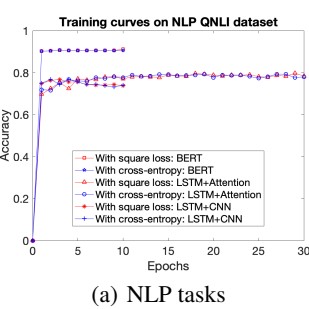
(a) NLP tasks

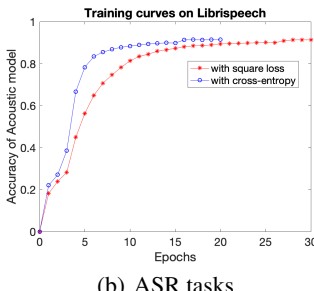
(b) ASR tasks

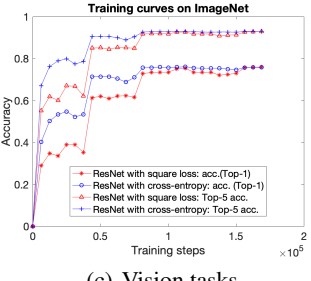
(c) Vision tasks

Figure 2: Training curves

We compare the convergence speed in terms of accuracy, and find that for 2-class NLP classification tasks, the training curves of training with the square loss and the cross-entropy are quite similar. Figure 2 (a) gives the accuracy of three model architectures trained with the square loss and the cross-entropy along different epochs for QNLI dataset. For all three models, BERT, LSTM+Attention, and LSTM+CNN, using the square loss converges as fast as cross-entropy loss, and achieves better/comparable accuracy to training with the cross-entropy.

**Convergence speed when class number is large** When the class number becomes large, as on speech dataset Librispeech and vision dataset ImageNet, training with the square loss may need more epochs to converge. Figure 2 (b) gives the classification accuracy of acoustic model along different epochs, and Figure 2 (c) gives the accuracy (Top-1) and Top-5 accuracy along different training steps of ResNet on ImageNet. Training with the square loss converges slower but reaches similar/better accuracy.

## 5 IMPLEMENTATION

We summarize the key points of implementation in this section. Full details and the exact parameters are given in Appendix B. Two important pieces of the implementation are (1) no softmax for training with the square loss and (2) loss rescaling for datasets with large number of classes.

**No softmax.** The widely accepted pipeline for modern neural classification tasks trained with the cross-entropy loss contains the last softmax layer before calculating the loss. When training with the square loss that layer needs to be removed as it appears to impede optimization.

**Loss rescaling mechanism.** For datasets with a small number of classes, we do not use any additional mechanisms. For datasets with a large number of output classes ($\geq 42$ in our experiments) we employ loss rescaling which helps to accelerate training. Let $(\boldsymbol{x}, \boldsymbol{y})$ denote a single labeled point, where $\boldsymbol{x} \in \mathbb{R}^d$ is the feature vector, and $\boldsymbol{y} \in \mathbb{R}^C$. Here $C$ is the number of output labels and $\boldsymbol{y} = [0, \ldots, \underbrace{1}_{c}, 0, \ldots, 0]$ is the corresponding one-hot encoding vector of the label $c$. We denote our model by $f : \mathbb{R}^d \to \mathbb{R}^C$.

Table 9: Rescaling parameters

| Dataset | #classes | k | M |
|---|---|---|---|
| MRPC | 2 | 1 | 1 |
| SST-2 | 2 | 1 | 1 |
| QNLI | 2 | 1 | 1 |
| QQP | 2 | 1 | 1 |
| TIMIT (CER) | 27 | 1 | 1 |
| TIMIT (WER) | 42 | 1 | 15 |
| WSJ | 52 | 1 | 15 |
| Librispeech | 1000 | 15 | 30 |
| MNIST | 10 | 1 | 1 |
| CIFAR-10 | 10 | 1 | 1 |
| ImageNet | 1000 | 15 | 30 |

The standard square loss for the one-hot encoded label vector can be written (at a single point) as

$$l = \frac{1}{C} \left( (f_c(\boldsymbol{x}) - 1)^2 + \sum_{i=1, i \neq c}^{C} f_i(\boldsymbol{x})^2 \right) \tag{1}$$

For a large number of classes, we use the rescaled square loss defined by two parameters, $k$ and $M$, as follows:

$$l_s = \frac{1}{C} \left( k * (f_c(\boldsymbol{x}) - M)^2 + \sum_{i=1, i \neq c}^{C} f_i(\boldsymbol{x})^2 \right).$$

The parameter $k$ rescales the loss value at the true label, while $M$ rescales the one-hot encoding (the one-hot vector is multiplied by $M$). Note that when $k = M = 1$, the rescaled square loss is same as the standard square loss in Eq. 1. The values of $k$ and $M$ for all experiments are given in Table 9. As in (Demirkaya et al., 2020), the parameter $k$ is used to increase the emphasis on the correct class in multiclass classification, and this paper proves how adding $k$ can simplify the optimization landscape. We find that for very large class numbers additional parameter $M$ further improves performance.

## 6 SUMMARY AND DISCUSSION

In this work we provided an empirical comparison of training with the cross-entropy and square loss functions for classification tasks in a range of datasets and architectures. We observe that the square loss outperforms cross-entropy across the majority of datasets and architectures, sometimes by a significant margin. No additional parameter modification except for adjusting the learning rate was necessary for most datasets. For datasets with a large number of classes (42 or more) we used additional loss rescaling to accelerate training. We note that all models used in our experiments were originally designed and tuned for training with the cross-entropy loss. We conjecture that if the neural architectures were selected and tuned for the square loss, performance would be further improved and no extra loss rescaling parameters would be necessary. Another important observation is that the final softmax layer, commonly used with cross-entropy, needs to be removed during training with the square loss.

While we could only explore a small sample of modern models and learning tasks, we believe that the scope of our experiments — ten different neural architectures and ten different datasets across three major application domains — is broad enough to be indicative of the wide spectrum of neural models and datasets. Our empirical results suggest amending best practices of deep learning to include training with square loss for classification problems on equal footing with cross-entropy or even as a preferred option. They also suggest that new theoretical analyses and intuitions need to be developed to understand the important question of training loss function selection.

## ACKNOWLEDGMENTS

The authors acknowledge support from NSF (IIS-1815697) and NIH (R01EB022899) and a Google Faculty Research Award. We thank Nvidia for the donation of GPUs and Google for the free access to the cloud TPUs provided by the TFRC program. LH thanks Wuwei Lan for helpful discussions on NLP experiments and Peidong Wang for discussions on ASR experiments. MB thanks his co-authors on (Muthukumar et al., 2020), D. Hsu, V. Multukumar, A. Narang, A. Sahai and V. Subramanian, for insightful discussions related to loss functions and the Simons Institute for the Theory of Computing, where the initial discussions took place. We thank Ryan Rifkin for valuable feedback.

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

## APPENDICES

## A    DATASETS AND TASKS

Below we provide a summary of datasets used in the experiments.

**NLP tasks**

- **MRPC** (Microsoft Research Paraphrase Corpus) (Dolan & Brockett, 2005) is a corpus of sentence pairs extracted from online news sources. Human annotation indicates whether the sentences in the pair are semantically equivalent. We report accuracy and F1 score.
- **SST-2** (The Stanford Sentiment Treebank) (Socher et al., 2013) is a task to determine the sentiment of a given sentence. This corpus contains sentences from movie reviews and their sentiment given by human annotations. We use only sentence-level labels, and predict positive or negative sentiment.
- **QNLI** is a converted dataset from the Stanford Question Answering Dataset (Rajpurkar et al., 2016) which consists of question-paragraph pairs. As in (Wang et al., 2018), this task is to predict whether the context sentence selected from the paragraph contains the answer to the question.
- **QQP** (Quora Question Pairs dataset) (Iyer et al., 2017) contains question pairs from the question-answering website Quora. Similar to MRPC, this task is to determine whether a pair of questions are semantically equivalent. We report accuracy and F1 score.

**ASR tasks**

- **TIMIT** (Garofolo et al., 1993) consists of speech from American English speakers, along with the corresponding phonemical and lexical transcription. It is widely used for acoustic-phonetic classification and ASR tasks. Its training set, validation set and test set are 3.2 hours, 0.15 hours, 0.15 hours long, respectively.

- **WSJ** (Wall Street Journal corpus) (Paul & Baker, 1992) contains read articles from the Wall Street Journal newspaper. Its training, validation and test set are 80 hours, 1.1 hours and 0.7 hours long, respectively.

- **Librispeech** (Panayotov et al., 2015) is a large-scale (1000 hours in total) corpus of 16 kHz English speech derived from audiobooks. We choose the subset train-clean-100 (100 hours) as our training data, dev-clean (2.8 hours) as our validation set and test-clean (2.8 hours) as our test set.

**Vision tasks**

- **MNIST** (LeCun et al., 1998) contains $60,000$ training images and $10,000$ testing $28 \times 28$ pixel images of hand-written digits. It is a 10-class image classification task.

- **CIFAR-10** (Krizhevsky & Hinton, 2009) consists of $50,000$ $32 \times 32$ pixel training images and $10,000$ $32 \times 32$ pixel test images in 10 different classes. It is a balanced dataset with $6,000$ images of each class.

- **ImageNet** (Russakovsky et al., 2015) is an image dataset with 1000 classes, and about 1.28 million images as training set. The sizes of its validation and test set are $50,000$ and $10,000$, respectively. All images we use are in $224 \times 224$ pixels.

# B  HYPER-PARAMETER SETTINGS

We give the implementation toolkits and specific hyper-parameter settings to help reproduce our results, and list the epochs needed for training with the square loss and the cross-entropy (CE) loss. The data processing is following the standard methods. For NLP tasks, it is the same as in (Wang et al., 2018), and for ASR tasks, it is the same as in (Watanabe et al., 2018). For vision tasks, we are following the default ones given in the implementation of the corresponding papers.

## B.1  HYPER-PARAMETERS FOR NLP TASKS

The implementation of BERT is based on the PyTorch toolkit (Wolf et al., 2019). The specific script we run is `https://github.com/huggingface/transformers/blob/master/examples/text-classification/run_glue.py`, and we use the bert-base-cased model for fine-tuning. LSTM+Attention and LSTM+CNN are implemented based on the toolkit released by (Lan & Xu, 2018). The specific hyper-parameters used in the experiments are in Table 10. As there are many hyper-parameters, we only list the key ones, and all other parameters are the default in the scripts.

Table 10: Hyper-parameters for NLP tasks

| Model | Task | Batch size | max_seq length | Learning rate w/ | | Epochs training w/ | |
|---|---|---|---|---|---|---|---|
| | | | | square loss | CE | square loss | CE |
| BERT | MRPC | 32 | 128 | 5e-5 | 2e-5 | 5 | 3 |
| | SST-2 | 32 | 128 | 2e-5 | 2e-5 | 3 | 3 |
| | QNLI | 32 | 128 | 2e-5 | 2e-5 | 3 | 3 |
| | QQP | 32 | 128 | 2e-5 | 2e-5 | 3 | 3 |
| LSTM+Attention | MRPC | 64 | 80 | 2e-4 | 1e-4 | 25 | 20 |
| | QNLI | 32 | $sent\_len^*$ | 1e-4 | 1e-4 | 20 | 20 |
| | QQP | 64 | 120 | 1e-4 | 1e-4 | 30 | 30 |
| LSTM+CNN | MRPC | 64 | 80 | 2e-4 | 1e-4 | 20 | 20 |
| | QNLI | 32 | $sent\_len^*$ | 8e-5 | 1e-4 | 20 | 20 |
| | QQP | 32 | 120 | 1e-3 | 1e-3 | 20 | 20 |

[*] The max sequence length equals the max sentence length of the training set.

## B.2  HYPER-PARAMETERS FOR ASR TASKS

The implementation of ASR tasks is based on the ESPnet (Watanabe et al., 2018) toolkit, and the specific code we use is the run.sh script under the base folder of each task, which is `https:`

`//github.com/espnet/espnet/tree/master/egs/?/asr1`, where '?' can be 'timit', 'wsj', and 'librispeech'. The specific hyper-parameters are following the ones in the configuration file of each task, which is under the base folder. We list the files which give the hyper-parameter settings for acoustic model training in Table 11.

Table 11: Hyper-parameters for ASR tasks

| Model | Task | Hyper-parameters | Epochs training w/ | |
|---|---|---|---|---|
| | | | square loss | CE |
| Attention+CTC | TIMIT | conf/train.yaml♮ | 20 | 20 |
| VGG+BLSTMP | WSJ* | conf/tuning/train_rnn.yaml | 15 | 15 |
| VGG+BLSTM | Librispeech | conf/tuning/train_rnn.yaml◇ | 30 | 20 |

* For WSJ, we use the language model given by `https://drive.google.com/open?id=1Az-4H25uwnEFa4lENc-EKiPaWXaijcJp`.  ♮ We set mtlalpha=0.3, batch-size=30.  ◇ We set elayers=4, as we use 100 hours training data.

### B.3 HYPER-PARAMETERS FOR VISION TASKS

The implementation of these models are based on the open source toolkits. For TCNN and EfficientNet, we use the open source implementation given by (Bai et al., 2018) and (Tan & Le, 2019), respectively. For Wide ResNet, we are based on the open source PyTorch implementation `https://github.com/xternalz/WideResNet-pytorch` (W-ResNet). For ResNet-50, our experiments are based on the Tensorflow toolkit `https://github.com/tensorflow/tpu/tree/master/models/official/resnet` (ResNet) implemented on TPU. The hyper-parameter settings for our vision experiments are in Table 12.

Table 12: Hyper-parameters for vision tasks

| Model | Task | Hyper-parameters | Epochs training w/ | |
|---|---|---|---|---|
| | | | square loss | CE |
| TCNN | MNIST♮ | the default in (Bai et al., 2018) | 20 | 20 |
| Wide-ResNet | CIFAR-10 | the default in W-ResNet, except wide-factor=20 | 200 | 200 |
| ResNet-50 | ImageNet | the default in ResNet, for square loss, learning rate=0.3 | 168885* | 112590* |
| EfficientNet | ImageNet | the default in EfficientNet-B0 of (Tan & Le, 2019) | 218949* | 218949* |

♮ We are doing the permuted MNIST task as in Bai et al. (2018).
* We give the training steps as in the original implementations.

## C EXPERIMENTAL RESULTS ON VALIDATION AND TRAINING SETS

Table 13: NLP results on validation set, accuracy

| Model | Task | train with square loss (%) | train with cross-entropy (%) | square loss w/ same epochs as CE (%) |
|---|---|---|---|---|
| BERT (Devlin et al., 2018) | MRPC | **85.3** | 85.0 | **85.3** |
| | SST-2 | 91.2 | **91.5** | 91.2 |
| | QNLI | **90.8** | 90.7 | **90.8** |
| | QQP | **90.8** | 90.7 | 90.6 |
| LSTM+Attention (Chen et al., 2017) | MRPC | **76.5** | 74.8 | 75.3 |
| | QNLI | **79.7** | **79.7** | **79.7** |
| | QQP | **86.0** | 85.5 | **86.0** |
| LSTM+CNN (He & Lin, 2016) | MRPC | **76.0** | 73.3 | **76.0** |
| | QNLI | **76.8** | **76.8** | **76.8** |
| | QQP | 84.0 | **85.3** | 84.0 |

Table 14: NLP results on validation set, F1 scores

| Model | Task | train with square loss (%) | train with cross-entropy (%) | square loss w/ same epochs as CE (%) |
|---|---|---|---|---|
| BERT | MRPC | 89.5 | **89.6** | 89.5 |
| (Devlin et al., 2018) | QQP | **87.5** | 87.4 | 87.4 |
| LSTM+Attention | MRPC | **83.7** | 83.3 | 83.5 |
| (Chen et al., 2017) | QQP | **82.1** | 81.7 | **82.1** |
| LSTM+CNN | MRPC | **82.6** | 81.4 | **82.6** |
| (He & Lin, 2016) | QQP | 77.4 | **80.2** | 77.4 |

We report the results for validation set of NLP tasks in Table 13 for accuracy and Table 14 for F1 scores.

The validation set results of the ASR tasks are in Table 15.

Table 15: ASR results on validation set, error rate

| Model | Task | train with square loss (%) | train with cross-entropy (%) | square loss w/ same epochs as CE (%) |
|---|---|---|---|---|
| Attention+CTC | TIMIT (PER) | **18.1** | 18.3 | **18.1** |
| (Kim et al., 2017) | TIMIT (CER) | **30.4** | 31.4 | **30.4** |
| VGG+BLSTMP | WSJ (WER) | **8.5** | 8.8 | **8.5** |
| (Moritz et al., 2019) | WSJ (CER) | **3.9** | 4.0 | **3.9** |
| VGG+BLSTM | Librispeech (WER) | **9.3** | 10.7 | 9.9 |
| (Moritz et al., 2019) | Librispeech (CER) | **9.4** | 11.1 | 10.2 |

We report the training result for NLP tasks in Table 16 for accuracy and F1 score in Table 17. The training results for ASR tasks and vision tasks are in Table 18 and Table 19, respectively.

Table 16: NLP results on training and test set, accuracy

| Model | Task | train with square loss (%) | | train with cross-entropy (%) | | square loss w/ same epochs as CE (%) | |
|---|---|---|---|---|---|---|---|
| | | Train | Test | Train | Test | Train | Test |
| | MRPC | 99.7 | 83.8 | 99.9 | 82.1 | 99.6 | 83.6 |
| BERT | SST-2 | 98.6 | 94.0 | 99.2 | 93.9 | 98.6 | 93.9 |
| (Devlin et al., 2018) | QNLI | 98.0 | 90.6 | 97.5 | 90.6 | 98.0 | 90.6 |
| | QQP | 96.2 | 88.9 | 98.0 | 88.9 | 96.2 | 88.8 |
| LSTM+Attention | MRPC | 94.6 | 71.7 | 84.9 | 70.9 | 93.2 | 71.5 |
| (Chen et al., 2017) | QNLI | 87.7 | 79.3 | 90.8 | 79.0 | 87.7 | 79.3 |
| | QQP | 93.7 | 83.4 | 91.5 | 83.1 | 93.7 | 83.4 |
| LSTM+CNN | MRPC | 98.3 | 73.2 | 92.5 | 69.4 | 98.3 | 72.5 |
| (He & Lin, 2016) | QNLI | 92.8 | 76.0 | 90.7 | 76.0 | 92.8 | 76.0 |
| | QQP | 91.3 | 84.3 | 95.7 | 84.4 | 91.3 | 84.3 |

Table 17: NLP results on training and test set, F1 scores

| Model | Task | train with square loss (%) | | train with cross-entropy (%) | | square loss w/ same epochs as CE (%) | |
|---|---|---|---|---|---|---|---|
| | | Train | Test | Train | Test | Train | Test |
| BERT | MRPC | 99.8 | 88.1 | 99.9 | 86.7 | 99.7 | 88.0 |
| (Devlin et al., 2018) | QQP | 94.5 | 70.9 | 97.2 | 70.7 | 94.5 | 70.7 |
| LSTM+Attention | MRPC | 96.1 | 80.9 | 89.5 | 80.6 | 94.7 | 80.7 |
| (Chen et al., 2017) | QQP | 91.9 | 62.6 | 89.2 | 62.3 | 91.9 | 62.6 |
| LSTM+CNN | MRPC | 98.8 | 81.0 | 94.5 | 78.2 | 98.8 | 81.0 |
| (He & Lin, 2016) | QQP | 88.0 | 60.3 | 94.2 | 60.5 | 88.0 | 60.3 |

Table 18: ASR results on training and test set, error rate

| Model | Task | train with square loss (%) | | train with cross-entropy (%) | | square loss w/ same epochs as CE (%) | |
|---|---|---|---|---|---|---|---|
| | | Train | Test | Train | Test | Train | Test |
| Attention+CTC | TIMIT (PER) | 0.9 | 20.8 | 4.8 | 20.8 | 0.9 | 20.8 |
| (Kim et al., 2017) | TIMIT (CER) | 4.5 | 32.5 | 11.6 | 33.4 | 4.5 | 32.5 |
| VGG+BLSTMP | WSJ (WER)* | 0.7 | 5.1 | 0.3 | 5.3 | 0.7 | 5.1 |
| (Moritz et al., 2019) | WSJ (CER)* | 0.3 | 2.4 | 0.1 | 2.5 | 0.3 | 2.4 |
| VGG+BLSTM | Librispeech (WER)* | 0.8 | 9.8 | 0.4 | 10.6 | 0.8 | 10.3 |
| (Moritz et al., 2019) | Librispeech (CER)* | 0.6 | 9.7 | 0.3 | 10.7 | 0.6 | 10.2 |

* For WSJ and Librispeech, we take 10% of the training set for the evaluation of the training error rate.

Table 19: Vision results on training and test set, accuracy

| Model | Task | train with square loss (%) | | train with cross-entropy (%) | | square loss w/ same epochs as CE (%) | |
|---|---|---|---|---|---|---|---|
| | | Train | Test | Train | Test | Train | Test |
| TCNN (Bai et al., 2018) | MNIST (acc.) | 98.3 | 97.7 | 99.5 | 97.7 | 98.3 | 97.7 |
| W-Resnet (Zagoruyko & Komodakis, 2016) | CIFAR-10 (acc.) | 100.0 | 95.9 | 100.0 | 96.3 | 100.0 | 95.9 |
| ResNet-50 | ImageNet (acc.) | 77.7 | 76.2 | 80.5 | 76.1 | 77.7 | 76.0 |
| (He et al., 2016) | ImageNet (Top-5 acc.) | 93.2 | 93.0 | 93.4 | 93.0 | 93.2 | 92.9 |
| EfficientNet | ImageNet (acc.) | 75.1 | 74.6 | 81.4 | 77.0 | 75.1 | 74.6 |
| (Tan & Le, 2019) | ImageNet (Top-5 acc.) | 93.0 | 92.7 | 94.0 | 93.3 | 93.0 | 92.7 |

# D  OUR RESULTS COMPARED WITH THE ORIGINAL WORK

We list our results for the models trained with the cross-entropy (CE) loss and compare them to the results reported in the literature or the toolkits in Table 20. As we observe, our results are comparable to the original reported results.

Table 20: Training with the cross-entropy loss, our results and the reported ones

| Model | Task | Our CE result | CE result in the literature |
|---|---|---|---|
| BERT* | MRPC (acc./F1) | 85.0/89.6 | 85.29/89.47 (Wolf et al., 2019) |
| | SST-2 (acc.) | 91.5 | 91.97 (Wolf et al., 2019) |
| | QNLI (acc.) | 90.7 | 87.46 (Wolf et al., 2019) |
| | QQP (acc./F1) | 90.7/87.4 | 88.40/84.31 (Wolf et al., 2019) |
| LSTM+Attention | | | N/A |
| LSTM+CNN | | | N/A |
| Attention+CTC | TIMIT (PER) | 20.7 | 20.5 (Watanabe et al., 2018) |
| | TIMIT (CER) | 32.7 | 33.7 (Watanabe et al., 2018) |
| VGG+BLSTMP | WSJ (WER) | 5.4 | 5.3 (Watanabe et al., 2018) |
| | WSJ (CER) | 2.6 | 2.4 (Watanabe et al., 2018) |
| VGG+BLSTM | Librispeech (WER) | 10.8 | N/A |
| | Librispeech (CER) | 11.0 | N/A |
| TCNN | MNIST (acc.) | 98.0 | 97.2 (Bai et al., 2018) |
| Wide-ResNet | CIFAR-10 (acc.) | 96.5 | 96.11 (Zagoruyko & Komodakis, 2016) |
| ResNet-50 | ImageNet (acc./Top-5 acc.) | 76.1/93.0 | 76.0/93.0 (Tan & Le, 2019) |
| EfficientNet | ImageNet (acc./Top-5 acc.) | 77.2/93.4 | 77.3/93.5 (Tan & Le, 2019) |

* The implementation in (Wolf et al., 2019) is using bert-base-uncased model, we are using bert-base-cased, which will result in a little difference. Also, as they didn't give test set results, here for BERT, we give the results of validation set.

The models marked with 'N/A' in Table 20 do not have comparable results reported in the literature. Specifically, LSTM+Attention and LSTM+CNN models for NLP tasks are implemented based on the toolkit released by (Lan & Xu, 2018), where they did not show results on MRPC and QNLI. The QQP results are not comparable with ours as they were using a different test set, while we are using the standard test set same as in (Wang et al., 2018). The VGG+BLSTM model for Librispeech dataset is based on ESPnet toolkit (Watanabe et al., 2018). Due to computational resources limitations, we only use train-clean-100 (100 hours) as training data and 1000 unigram based dictionary for acoustic model training, while they use 1000 hours of training data with at least 2000 unigram dictionary.

# E    REGULARIZATION TERMS

We give the regularization term of each task in Table 21. 0 means we didn't add regularization term. For WSJ, check the details at line 306 of `https://github.com/espnet/espnet/blob/master/espnet/nets/pytorch_backend/rnn/decoders.py`.

Table 21: Regularization term for each task

| Model | Task | dropout* | batch norm | Regularization Term |
|---|---|---|---|---|
| BERT | MRPC/SST-2/QNLI/QQP | 0.1 | N | 0 |
| LSTM+Attention | MRPC/QNLI/QQP | 0.5 | N | 0 |
| LSTM+CNN | MRPC/QNLI/QQP | 0.0 | N | 0 |
| Attention+CTC | TIMIT | 0.0 | N | 0 |
| VGG+BLSTMP | WSJ | 0.0 | N | label smoothing based |
| VGG+BLSTM | Librispeech | 0.0 | N | 0 |
| TCN | MNIST | 0.05 | N | 0 |
| Wide-ResNet | CIFAR-10 | 0.0 | N | 0 |
| ResNet-50 | ImageNet | 0.0 | Y | $\frac{10^{-4}}{2}\sum_{i=1}^{n} \boldsymbol{w_i^2}$ |
| EfficientNet | ImageNet | 0.0 | Y | $\frac{10^{-5}}{2}\sum_{i=1}^{n} \boldsymbol{w_i^2}$ |

\* For dropout, 0.0 means have not apply dropout.

# F    VARIANCE OF ACCURACY AMONG DIFFERENT RANDOM SEEDS

Figure 3 gives the error bar of 5 runs corresponding to 5 different random seeds, along with the results for each inidividual run. In the left of each subfigure is the result of training with the square loss, while in the right is result of the cross-entropy. As can be seen in Figure 3, using the square loss has better accuray/error rate and smaller variance in NLP and ASR tasks, which indicates that training with the square loss for those classification tasks is statistically better.

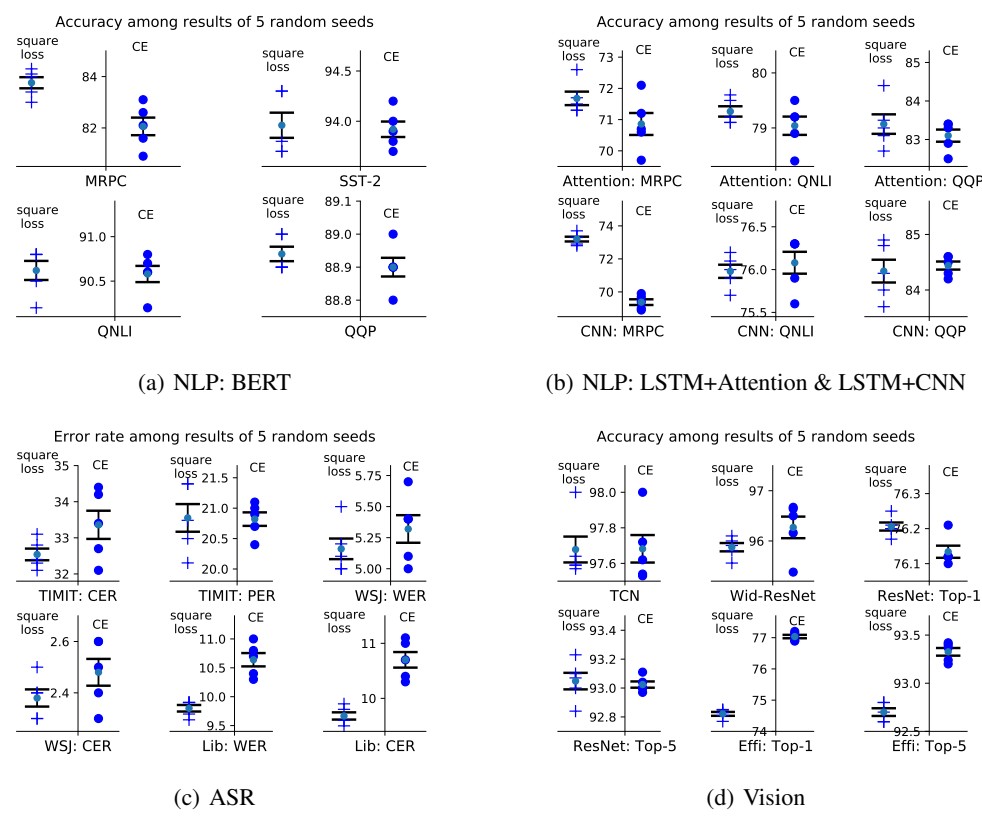

(a) NLP: BERT

(b) NLP: LSTM+Attention & LSTM+CNN

(c) ASR

(d) Vision

Figure 3: Accuracy/error rate variance of results among 5 random seeds

