# OpenReview forum: "EVALUATION OF NEURAL ARCHITECTURES TRAINED WITH SQUARE LOSS VS CROSS-ENTROPY IN CLASSIFICATION TASKS"
_ICLR.cc/2021/Conference — ICLR 2021 Poster_

### Official Review · AnonReviewer4 · 2020-10-26
**Interesting paper that challenges the conventional wisdom of CE loss being superior to MSE loss in clasffication tasks.**

**Rating:** 8
**Confidence:** 5

**Review:**


I think this a very good contribution to ICLR given the topic and the quality of the submission (originality, contribution to the stare of the art, experimental evidence, etc)

 Some of the strong points of the submission are summarized as follows:

1.	The paper tackles a very interesting subject, questioning the conventional wisdom the CE is superior to MSE loss in a wide range of machine learning problems.
2.	Very good introduction and motivations sections. The hypothesis, as well as the main motivations are discussed succinctly but in a very logical manner including historical aspects leading to the current state of affairs (in terms of the manner in which models are trained) that might be helpful for interested readers not sufficiently familiar with the aspects discussed in the article.
3.	The state of the art (despite the previous comment) contextualizes the subject matter in a succinct but comprehensive manner.
4.	I have read people making similar claims in other forums and articles, but the authors here provide a very thorough and careful experimental design, which helps to validate their hypothesis. However, it would be interesting to see how these experiments generalize to problems (in particular in computer vision) where datasets are noisier or where the image quality/resolution are lower.
5.	The experimental design is good, showing a careful analysis to validate the proposal and several ablation studies to confirm that the hypothesis holds for various machine learning domains, as well as several datasets.
6.	The foundations for the method are presented in great detail in a formalized manner and provides sufficient to assess the validity of the proposed experimental design.

However, there are certain things that in my opinion could be improved:

1.	Future work could be further elaborated and discussion in specific domains (medical imaging, for instance) could be further discussed.

---

> ### Author Response · Authors · 2020-11-19
> **Response to Reviewer 4**
>
> Thanks for the encouraging comments!
>
> We agree that understanding the choice of loss functions in different domains is very important. Indeed, our results already hint that there may be systematic differences between domains. However, we feel the evidence is not yet strong enough and  significantly more experimentation is needed before definite conclusions can be made.  This is an important direction of future work. We will adjust the conclusion to reflect this.

---

### Official Review · AnonReviewer3 · 2020-10-28
**Empirical Evaluation of Cross Entropy vs Square Loss**

**Rating:** 6
**Confidence:** 4

**Review:**

The paper compares cross-entropy and squared losses on a wide range of tasks. The focus primarily is on thorough empirical evaluation of these two losses on NLP, Vision and Speech tasks. The paper shows that squared loss is better or competitive with cross-entropy loss in most cases. Most of the experiments and comparisons seem to be well done; parameters, setups etc. are well explained.

I believe a few additional tasks and settings would have helped put a better picture of the comparison.
For speech, the application of the two losses in tasks beyond ASR might give more insights. Similar for vision, classification tasks other than image classification (on MNIST/CIFAR and Imagenet) might be useful.

Additional learning settings might also be useful. For example, there is considerable amount of work (e.g R1 and R2 below) on noisy label learning, where the loss function (often cross entropy losses) is modified for noisy label condition. What do we expect for these two losses  in this noisy label setting ? Essentially what can we expect in learning settings beyond the supervised training in vanilla form.

For square loss, scaling is done for a large number of classes. The motivation behind it is not very clear. Also, it is perhaps worth showing results for squared loss with softmax outputs. Especially for the large number of classes. Can softmax be advantageous in this case for performance, even though computationally it will be worse than the scaling mechanism applied.

R1. Generalized Cross Entropy Loss for Training Deep Neural Networks with Noisy Labels
R2. Normalized Loss Functions for Deep Learning with Noisy Labels

---

> ### Author Response · Authors · 2020-11-19
> **Response to Reviewer 3**
>
> Thanks for the comments!
>
> --> For the comment on additional learning settings about noisy label learning
>
>   We agree that the noisy label setting is important and interesting to explore. However we feel that it goes beyond the scope of this paper, as it requires significant additional experimental analysis in different settings (e.g., understanding the results depending on the noise level or types of noise). This is a direction of future work.
>
> --> For the comment about square loss with softmax outputs
>
>   We have done experiments with the square loss with softmax output, however the results  generally appeared to be poor. For example, for ImageNet dataset generalization performance for square loss with softmax outputs was close to a random guess.

---

### Official Review · AnonReviewer2 · 2020-10-28
**Important work towards understanding loss function choice**

**Rating:** 7
**Confidence:** 4

**Review:**

Summary: This paper compares the more popular cross entropy loss function to the squared loss function for classification tasks. The authors look at NLP, ASR and and CV tasks, keeping the architecture the same (as much as possible) and varying the loss functions. The authors demonstrate that although cross entropy is more commonly used in state-of-the-art architectures and is standard across tutorial code, square loss functions are often times advantageous.

Strengths:
This paper highlights an ongoing issue in deep learning -- we often take for granted what 'best practices' are and do not sufficiently investigate the best loss function, etc. when starting a new problem.

The study covers a reasonable amount of types of problems and architectures and conducts 5 random initializations which is rare in many ML papers.

Weaknesses:
While this work is highly important and the study was well-done, the novelty is a little low. Many papers have done variations of this same type of work to answer this question.

While I understand there are only so many experiments that a group can do, there are some limitations to this study -- including not performing hyper-parameterization for architectures comparing the two loss functions which may reveal some important distinctions and use-cases.

A minor comment is that the average accuracies are reported but not standard deviation to get a sense of the variation across performance for both loss functions. Statistical significance calculations would also be helpful to interpret results.

Question for authors:
What other areas do you find people are using the wrong hyper-parameter choice? This could be an interesting discussion write-up for researchers to re-consider how they select their training paradigm.

I recommend accepting this work. The study was well-done for answering this question and more thorough than related work. My reason for not giving a higher score is that the work is not highly novel.

---

> ### Author Response · Authors · 2020-11-19
> **Response to Reviewer 2**
>
> Thanks for the comments!
>
> --> “the average accuracies are reported but not standard deviation to get a sense of the variation across performance for both loss functions. Statistical significance calculations would also be helpful to interpret results.”
>
>   We give the standard deviations (among 5 runs with different initializations) in Table 8. Also we plot the error bars in Figure 1 and Figure 3. Figure 1 shows the difference between accuracy (or error rate) between square loss and CE for each initialization, while Figure 3 shows the error bars of 5 runs corresponding to 5 different initializations.

---

### Official Review · AnonReviewer1 · 2020-11-01
**The paper shows that the square loss is usually as effective as the cross-entropy loss across classification tasks in a variety of domains and model architectures.**

**Rating:** 7
**Confidence:** 3

**Review:**

This paper questions the omnipresence of cross-entropy loss for classification tasks while showing that square loss also yields competitive results. The experimental section spans a wide variety of tasks and architectures covering NLP, ASR, and Vision. The authors find that the model performance is more stable w.r.t. the random initialization of parameters when trained using the square loss. For NLP and ASR datasets, the authors find that the square loss yields slightly better results. The authors also report that the square loss usually takes more time to converge and requires rescaling with a larger number of classes.
The authors do not dwell on the fundamental reasons behind the observations made in this paper. However, I believe that these observations are indeed useful to advance further research for better loss functions.

Strong Points:
1. Diverse experiments and insightful results

Weak Points:
1. The paper does not make an effort to provide well-grounded explanations for the experimental observations.
2. It is unclear how the CTC-based architectures for ASR were modified for square-loss. An explanation in the paper would be helpful.

Questions:
 1. How were the CTC-based architectures for ASR were modified for square-loss. ?
 2. What is the opinion of authors about the trade-off between the hyperparameter associated with the loss-scaling Vs. the simplicity of the cross-entropy loss for a larger number of classes?

---

> ### Author Response · Authors · 2020-11-19
> **Response to Reviewer 1**
>
> Thanks for the comments!
>
> --> “The paper does not make an effort to provide well-grounded explanations for the experimental observations.”
>
>   Currently there is  limited theoretical understanding of the choice of loss functions in modern neural architectures or even for simpler settings, such as kernel machines. Thus we concentrated on identifying and clearly demonstrating these practically important phenomena with the hope that future work will shed light on their theoretical aspects.
>
> --> “It is unclear how the CTC-based architectures for ASR were modified for square-loss. An explanation in the paper would be helpful.”
>
>   Only the TIMIT experiments have CTC-based architectures and they use both CTC-loss and CE (a weighted sum of the two losses) in the original implementation. We only replaced the CE part with the square loss, and didn’t change the CTC part. We updated the paper to make this clear, thanks for pointing it out.
>
> --> For Question 2: “What is the opinion of authors about the trade-off between the hyperparameter associated with the loss-scaling Vs. the simplicity of the cross-entropy loss for a larger number of classes?”
>
>    This is a very interesting question, and we have following points:
>       For tasks with small class numbers, rescaling for the square loss is not necessary, so there is no trade-off.
>       For tasks with a large number of classes, rescaling the loss function was needed to obtain competitive results. However, it is not clear whether it is a true trade-off or simply the result of our insufficient understanding of the underlying problem.  We conjecture that better theoretical understanding of the problem can give us a simple rescaling rule based on the number of classes and no additional hyper-parameter tuning will be needed. This is an important future direction for theoretical and empirical analysis.

---

### Public Comment · ~Didier_Chételat1 · 2020-11-12
**Why are the two losses not equivalent?**

I would like to thank the authors for proposing this interesting work that support their claims with very extensive experiments. I had a more theoretical question regarding their proposal.

Consider for simplicity a binary classification problem with f_0(x), f_1(x) be the outputs of the neural network. Say we want to train with a squared error loss, and as mentioned on page 8 of the paper, let there be no final softmax layer. The loss then is
$L = (f_0 - [1-y])^2/2 + (f_1-y)^2/2.$

Now, say we would like to give a probabilistic interpretation to this. Minimizing the cross-entropy loss is the same as doing maximum likelihood on a Bernoulli response Y where the probabilities would be given by $\text{softmax}([f_0, f_1])$. Thus I was wondering, if this loss was the result of doing maximum likelihood on a probabilistic model, what would that probabilistic model be?

Working out the math, it would mean that $P[Y=y|X=x] \propto \exp(-(f_0 - [1-y])^2/2 - (f_1-y)^2/2)$, which is  $\exp(-f_0^2/2 + [1-y]f_0 - [1-y]/2 - f_1^2/2 + yf_1 - y/2)$ because $y^2=y$ and $(1-y)^2=1-y$. Thus $P[Y=y|X=x] \propto \exp([1-y]f_0 + yf_1) = \exp(f_y(x))$, which is to say

$P[Y=y|X=x] = \frac{\exp(f_y(x))}{\exp(f_0(x))+\exp(f_1(x))}.$

Assuming I have not made a mistake, this would mean that Y|X=x would have exactly the same implied distribution as if you had used the cross-entropy loss. Thus any differences between using the cross-entropy loss or the squared error loss would not come from differences in modeling the relationship between the response and the covariates.

Looking at experimental results, the differences between cross-entropy and squared-error loss seem very small (tables 2, 3, 5, 7). In fact, it is not clear to me whether the differences are even statistically significant. Thus I am wondering, could you expand as to why, despite the argument above, these two losses are not in fact equivalent?

---

> ### Author Response · Authors · 2020-11-19
> **They are not equivalent because the minimizer of the two losses are different**
>
> Thanks for the comments!
>
> Perhaps the easiest way to see that the two losses are not equivalent is to consider the case with an infinite number of y’s and a single fixed x.  Assuming you have two outcomes 1 with probability p and -1 with probability 1-p, the minimizer of the square loss is 2*p -1, while the minimizer of the cross-entropy loss is log(p/(1-p)). It is true that both are admissible loss functions (as is the hinge loss) -- the corresponding prediction rules (the sign of the minimizer) coincide with the Bayes optimal predictor, which is 1 if p>0.5 and -1 otherwise.  However, there is generally no reason (as is clear from the fact that the numerical values of the losses are different) that ERM algorithms that optimize different loss functions will produce same (or even similar) predictors.

---

### Decision · Program_Chairs · 2021-01-07
**Final Decision**

**Decision:**

Accept (Poster)

**Comment:**

The paper questions the use of cross-entropy loss for classification tasks and shows that using squared error loss can work just as well for deep neural networks. The authors conduct extensive experiments across ASR, NLP, and CV tasks. Comparing cross-entropy to squared error loss is certainly not novel, but the conclusions of the paper, backed by a lot of experimental evidence, are certainly thought-provoking.

I would have liked to see a bit more analysis into the results of the paper, and perhaps a bit more theoretical justification. That said, the paper will be of interest to the community, given the ubiquity of classification tasks.